# *Helicobacter pylori* Outer Membrane Proteins and Virulence Factors: Potential Targets for Novel Therapies and Vaccines

**DOI:** 10.3390/pathogens13050392

**Published:** 2024-05-08

**Authors:** Zahra Sedarat, Andrew W. Taylor-Robinson

**Affiliations:** 1Cellular & Molecular Research Centre, Shahrekord University of Medical Sciences, Shahrekord 8813833435, Iran; sedaratzahra@gmail.com; 2College of Health Sciences, VinUniversity, Gia Lam District, Hanoi 67000, Vietnam; 3Center for Global Health, Perelman School of Medicine, University of Pennsylvania, Philadelphia, PA 1904, USA

**Keywords:** *Helicobacter pylori*, outer membrane protein, virulence factor, gastric cancer, gastric disease

## Abstract

*Helicobacter pylori* is a gastric oncopathogen that infects over half of the world’s human population. It is a Gram-negative, microaerophilic, helix-shaped bacterium that is equipped with flagella, which provide high motility. Colonization of the stomach is asymptomatic in up to 90% of people but is a recognized risk factor for developing various gastric disorders such as gastric ulcers, gastric cancer and gastritis. Invasion of the human stomach occurs via numerous virulence factors such as CagA and VacA. Similarly, outer membrane proteins (OMPs) play an important role in *H. pylori* pathogenicity as a means to adapt to the epithelial environment and thereby facilitate infection. While some OMPs are porins, others are adhesins. The epithelial cell receptors SabA, BabA, AlpA, OipA, HopQ and HopZ have been extensively researched to evaluate their epidemiology, structure, role and genes. Moreover, numerous studies have been performed to seek to understand the complex relationship between these factors and gastric diseases. Associations exist between different *H. pylori* virulence factors, the co-expression of which appears to boost the pathogenicity of the bacterium. Improved knowledge of OMPs is a major step towards combatting this global disease. Here, we provide a current overview of different *H. pylori* OMPs and discuss their pathogenicity, epidemiology and correlation with various gastric diseases.

## 1. Introduction

*Helicobacter pylori* is considered an ancient microorganism, the existence of which can be traced back to before the voyages of Christopher Columbus [1]. Yet, it took until the early 1980s for the bacterium to be identified by the Australian physicians Barry Marshall and Robin Warren. For discovering *H. pylori* as the principal cause of gastritis and peptic ulcer disease and mucosa-associated lymphoid-tissue (MALT) lymphoma [2,3], they were awarded the Nobel Prize in Physiology or Medicine in 2005. Chronic *H. pylori* infection is a predisposing factor for a range of other health conditions including ischemic stroke, Alzheimer’s disease, multiple sclerosis, autoimmune neutropenia, vitamin B12 deficiency, diabetes mellitus, cholelithiasis, idiopathic thrombocytopenic purpura, iron-deficiency anemia, cardiovascular diseases, hepatobiliary diseases, and biofilm-related infections, although further research is needed to verify each proposed link [4,5,6,7,8,9,10,11,12,13,14,15]. It is estimated that more than half of the world’s population is infected with this microorganism, its prevalence in developing countries reaching 70–90%, compared to developed nations where it is between 20% and 30% [16,17]. Typically, a person becomes infected with *H. pylori* during childhood through oral–fecal or oral–oral transmission [18]. This Gram-negative, helical bacterium is a major source of global gastric cancer mortality, so it is considered as an oncogenic pathogen (oncopathogen) and hence is classified as a class I carcinogen by the World Health Organization [19]. It is equipped with different virulence factors including flagella, lipopolysaccharide (LPS), urease, and outer membrane proteins (OMPs), which are encoded by many paralogous gene families. It owes its characteristically high motility to its 4–6 co-located flagella, which facilitate its movement and colonization of the stomach mucosa layer. Urease production provides ammonia for bacterial protein synthesis and neutralizes gastric acid, thereby making the stomach a preferred environment for colonization. This factor can damage host tissue via several mechanisms, which, together with the inflammatory immune response that this triggers, causes ulceration. Similarly, the unique structure of LPS promotes bacterial pathogenicity by facilitating attachment to gastric mucosa, thus supporting persistence of infection [20,21,22,23].

It is estimated that only approximately 20% of *H. pylori* carriers develop symptoms of disease. Chronic gastritis is a condition ascribed for *H. pylori* carriers without any clinical symptoms. At the same time, this pathogen is a risk factor for progression to gastric problems like a peptic ulcer [24,25,26]. Chronic gastritis follows colonization of the stomach by *H. pylori*, which resists clearance and causes mucosal inflammation and atrophy. Peptic ulcer formation, a consequence of damaged mucosa through stomach acid activity, is accelerated by the chronically acidic environment [27]. These sores can develop either into a lesion inside the stomach, known as a gastric ulcer, or inside the adjoining duodenum within the small intestine, termed a duodenal ulcer [28]. Importantly, having chronic gastritis increases a person’s risk of acquiring severe gastric conditions, notably gastric cancer that most often manifests as stomach adenocarcinoma [29].

An array of contributing factors, such as genetic susceptibility, diet, environmental variables, smoking and physical activity, are involved in progression to severe stomach conditions [30]. Studies showed that *H. pylori* is the leading cause of 63.4% of all stomach cancer and 75% of non-cardia gastric cancer (that affects the first part of the stomach) [31]. While there is now a decreasing trend in the rate of gastric cancer worldwide, it is still the second highest cause of cancer mortality [32]. In order to eradicate *H. pylori*, antibiotic therapy is suggested for gastric disorders. Currently, antibiotics like clarithromycin, amoxicillin or metronidazole are used in combination with proton pump inhibitors as a standard treatment [33]. It should also be noted that eradicating this microorganism may provoke some extra-gastric diseases, in particular iron deficiency, idiopathic thrombocytopenic purpura, chronic idiopathic urticaria and anemia. Further studies are required to confirm this correlation [34].

The first step for *H. pylori* to induce inflammation and cause infection is to colonize and attach to gastric mucosa. Usually, this happens through OMPs which play a pivotal role in adherence and pathogenicity. To date, there are approximately 64 members of this family which are recognized [35,36,37]. Also, five paralogous genes of OMPs have been identified. Through analyzing strains of *H. pylori*, 26,695 and J99, many OMPs were identified. In one study, five family members of OMP, each with its own sub-family, were recognized. These families include the major OMPs Hop, Hor, Hof and Hom, iron-regulated OMPs, FecA/FrpB-like proteins, and efflux pump OMPs (Table 1) [37].

Here, we offer a contemporary perspective on *H. pylori* OMPs and virulence factors like VacA and CagA. We discuss their roles in pathogenicity, epidemiology, and correlation with gastric conditions. Additionally, we delve into therapeutic approaches targeting *H. pylori* OMPs and virulence factors, and highlight challenges to vaccine development.

## 2. *Helicobacter pylori* Virulence Factors

### 2.1. Cag A and Vac A

**Definition, diversity, classification and their significance:** Vacuolating cytotoxin, or VacA, and Cag (cytotoxin-associated genes) pathogenicity island (PAI), encoding a bacterial type IV secretory apparatus (T4SSs), are two main factors involved in *H. pylori* pathogenicity (Figure 1). CagA is a protein of 116–140 kDa molecular weight that is expressed by almost 70% of strains and which produces a specific cytotoxin [38]. The significant role of this protein in *H. pylori*-infected patients has led to the isolates being defined as belonging to one of two groups, either CagA-positive (type I) or CagA-negative (type II). Epithelial cells and cells of the immune system are considered as two main targets for VacA, in which is expressed by all *H. pylori* strains [39,40]. VacA protein has cytotoxic activity that is due to its ability to drive intracellular vacuolization [41]. It has been demonstrated that various cell types are vulnerable to this toxin [40]. It has escape mechanisms to avoid the highly acidic environment of the stomach [42]. Different receptors are recognized for VacA, yet their roles and importance are not clear [43]. Similar to CagA, this virulence factor is expressed only in *H. pylori* type I [44]. Notably, there are three types of VacA genotype predicated on their signal sequence, namely *s1a*, *s1b* and *s2*, as well as *m1* and *m2*, which is based on middle-region alleles of the *vacA* gene [45]. Regarding *vacA* allelic diversity, regions including *s*-region (signal), *m*-region (middle), *i*-region (intermediate), *d*-region (deletion) and *c*-regions are elucidated. Based on the deletion at the 3′ end of the *vacA* gene, different types are investigated. The *i* region exists in three types (*i1*, *i2* and *i3*), while all other regions are classified into two types (*s1*, *s2*, *m1*, *m2*, *c1*, *c2*, *d1* and *d2*). More variants within these regions are proposed, of which K, E and Q-types are conspicuous [46].

The *cag* PAI is a 40 kb DNA sequence as that encodes type IV secretion system (T4SS) and CagA protein. This generates a pilus via which the bacterium can inject CagA protein into a host cell [47,48]. There are twelve recognized components of T4SSs in Gram-negative bacteria, including VirD4 and VirB1-11. It is organized into three parts: outer membrane core complex, inner membrane complex, and extracellular pilus [49,50]. Upon delivery of CagA into the cell and phosphorylation of a C-terminal EPIYA motif, the signaling pathway is activated via binding of CagA to the SH2 domain. Host cell changes occur after components interact with both phosphorylated and non-phosphorylated CagA. Of note are changes in cell junction, elongation, polarity, proliferation and proinflammatory response [51,52]. Various bacterial proteins such as CagM, CagX, CagY, CagT and Cag3 that form a part of CagPIA are encoded by a 41 nm long core structure. Among these, CagX and CagY are associated with the T4SS channel [53]. An interaction between CagL on the T4SS and α5β1 integrin leads to CagA transposition and pilus formation. Consequently, cells become more irregular as a result of phosphorylation at the 3′ end of *CagA* gene (EPIYA), which is located in the PAI [44,54]. CagA is a highly immunogenic protein that comprises two types, CagAI and CagAII on the right or left segment, respectively [38,44]. This *H. pylori* type I virulence factor is linked to gastroduodenal disease and its gene may be acquired horizontally [44,55]. Upon bacterial attachment and infection, CagA will activate signal factors such as interleukin (IL)-8, which depend on the Cag PAI activity [44,56]. IL-8 and NFκB will pave the way for inflammation and carcinogenesis [54]. Another gene called *cagE*, located in the *cagI* and in proximity to *cagA*, has similarity with *ptlC* in *Bordetella pertussis* [44]. This gene is considered a better marker of pathogenicity, although further verification is needed [57]. In addition, there is a correlation among these virulence factors and other OMPs including IceA, BabA, HopQ, OipA, SabA and HopZ [58].

**Geographical variances and clinical associations:** Several studies have investigated associations between these two antigens and different gastric conditions, yet neither is considered as an indicator of gastric cancer [59,60]. A high frequency of CagA-positive isolates in patients with gastric cancer was reported [61]. Different results obtained vary by geographical region. In one study performed in various countries, *s1c-m1*, *s1b-m1* and *s1a-m1* of *vacA* were the predominant genotypes in Japan and Korea, US, and Colombia, respectively. Although *cagA* genotype was predominant in all nations, no relationships with clinical outcomes were identified [59]. In Egypt, however, 68.7% of patients with a gastric ulcer, 50% of gastric carcinoma patients and 33.3% of gastritis cases were positive for *cagA* gene expression [62]. On the other hand, in an Australian cohort of *H. pylori*-infected individuals, 78% and 85% of cases of duodenal ulcer and gastric cancer, respectively, were positive for the *cagA* gene [63]. Another study showed an association between *vacA s1a*, *cagE* and *cagA* with gastric cancer and duodenal ulceration [64]. Additionally, a correlation between *d*-region and gastric atrophy and neutrophil infiltration was reported. There is a close relationship between geographical region and distribution of VacA subtypes. It is apparent that *s1*/*m1* and *i1* are predominant genotypes in northeast Asia. Also, a close relationship between VacA subtypes and gastric disorders is demonstrable. Furthermore, an association between *s1a*, *s1c* and *m1* with gastric cancer, peptic ulcer and intestinal metaplasia was reported [3,65,66,67].

### 2.2. H. pylori Outer Membrane Proteins

Hop is the largest family of *H. pylori* OMPs, with 32 known members, yet they are collectively encoded by only 4% of the bacterial genome [36,37]. Hop A-E act as porin proteins as well as a channel through which antimicrobial agents permeate into the cell. Hence, many of them are potential candidates for development vaccine [68,69]. This group contains two divisions, Hop and Hor proteins. Interestingly, members of the latter lack a hop motif but still have an N-terminal motif, as do Hop proteins, and which is greatly variable in size. The former is divided into two groups based on the C-terminus [37].

Adhesion to host epithelial cells is the very first step for *H. pylori* colonization and persistence, which is mostly mediated by OMPs and T4SS [70]. There are three distinct steps of infection: colonization; attack of the gastric mucosa; and escape from the immune system. Attachment to mucins depends on several variables including type of mucin, anatomical site, pH, *H. pylori* strain and gastritis status. Also, interaction between *H. pylori* and host Lewis antigens, Le^a,b,x,y^, attributed to Hop proteins such as SabA and BabA, is vital to this process [71,72].

Protected by a mucus layer and composed mostly of MUC5AC and MUC6, the gastric epithelium is responsible for a glycosylation pattern that varies between gastric disorders. MUC2 is a type of mucin that does not exist in normal mucosa but instead is found mostly in intestinal metaplasia in which goblet cells are predominant. Understanding more about mucin expression patterns is important as *H. pylori* adhesion is mediated through interaction between these antigens and virulence factors [73,74,75].

#### 2.2.1. Hop B and Hop C

HopB and HopC, also known as AlpA and AlpB, are encoded by the *alp A/B* locus (OMP 20 and 21, respectively). Homology of AlpA/B among various *H. pylori* strains is reported as more than 90%. While the role of these proteins remains to be substantiated, they are assumed to be involved in adhesion [37,76,77], for which laminin serves as a receptor. Any interruption to Hop B/C leads to diminished binding of *H. pylori* to laminin [78]. In addition, these proteins are responsible for producing cytokines such as IL-6, IL-8 and for activating signal transduction [76,79,80]. Gastric damage and modulation of cell signaling are consequent to AlpA/B adhesion [81]. Both play a key role in *H. pylori* colonization, although HopB appears to be more important [82]. New insights into the molecular mechanism of HopC indicate a function in biofilm formation. As described later, *H. pylori* can construct biofilm in human gastric cells, HopC being one of the OMPs with the capability to contribute to this in outer membrane vesicles (OMVs) [83].

Regarding the pathogenicity of HopB/C, there is insufficient information correlating their presence with clinical outcomes. Analysis of 200 *H. pylori* isolates revealed that all express these proteins, which suggests their important roles [80]. Interestingly, in another study severe gastric symptoms were associated with some *H. pylori* virulence factors such as HopB and VacA, with a high prevalence of HopB in cases of gastric cancer and peptic ulcers (>80%), implying the importance of this OMP to predictions of infection outcome [67].

#### 2.2.2. Hop H, a Phase-Variable Protein

HopH, originally identified as outer inflammatory protein or Oip A (Hpo638), is a phase-variable protein, the alleles of which are present in almost all *H. pylori* strains. A high rate of diversity within CT dinucleotide repeats occurs in the *oipA* gene. Similar to other OMPs, it is assumed that HopH is involved in epithelial cell adhesion, although there are discrepancies arising from diversity between strains. This protein can also induce IL-8 production, cell-signaling and toxic events, as well as apoptosis [84,85,86]. These properties are independent of Cag PIA activity. This means that those strains which contain both virulence factors are capable of producing, for instance, higher levels of IL-8 [87]. Both functional and non-functional types of OipA are known [87,88]. Interestingly, an association between this protein and other virulence factors such as CagA, VacAs1 and BabA has been demonstrated [58,89,90].

**Hop H association with clinical outcomes:** A correlation between the presence of HopH and gastric disorders such as gastric cancer and peptic ulceration has been established. A study in which several virulence factors were examined together showed that gene expression could be a useful predictor of progression to gastric cancer in patients with precancerous gastric lesions, although paradoxical findings have raised doubts [78,91,92,93]. An investigation of *hopH* gene polymorphism led to two proposals for its pathogenicity, enhanced bacterial adhesion and correlation with the presence of other virulence factors [94]. In another study performed on gastritis and peptic ulcers, a high prevalence of the *oipA* gene was reported, which could imply a relationship between this gene and disease progression [95]. Similarly, a study performed on patients with gastritis, gastric carcinoma or duodenal ulcers showed an association with virulence factors such as CagA, VacA, IceA, BabA and OipA. However, only OipA was recognized as a distinctive factor for clinical outcomes. Nonetheless, as this factor is common among patients, it should be applied as a predictor only in combination with other virulence factors [88]. Several trials reported a connection between CagA and OipA expression in which slipped strand mispairing of complementary bases during DNA replication enhances bacterial adaptability. Conversely, OipA was reported as a non-significant marker in one study which used PCR to detect and differentiate *H. pylori* virulence factors and to predict clinical outcomes [67].

#### 2.2.3. Hop P

This protein is also known as sialic acid-binding adhesin or SabA. for which the human Lewis (Le) histo-blood group antigens Le^x^ and Le^a^ are the main receptors. Sialyl-dimeric-Lewis x glycosphingolipid, defined as *H. pylori* receptor, is overexpressed in the stomach of infected people as *sLe^x^* and *sLe^a^* gene expression is upregulated during inflammation. In contrast, in the gastric mucosa of healthy people sialylated glycoconjugates are not abundant [96,97,98,99,100]. Other receptors for SabA have been identified. It can bind to α2-3-linked sialic acids and other sialic acid receptors [101], while laminin in the extracellular matrix also serves as a receptor [102]. *H. pylori* can bind specifically to glycosylated mucins, located in the proximity of epithelial cells, which helps it to maintain long-term infection [103]. Additionally, the polymorphic nature of *H. pylori* is attributed to SabA binding to sialylated carbohydrates. This is a unique strategy of adaptation for *H. pylori* [104], which tends to colonize those stomach areas with low acidity and high levels of HopP receptors [105].

SabA is classified as a protein that is regulated by phase variation. This means that *H. pylori* can switch expression of the gene on or off depending on circumstances [106]. Interestingly, *sabA* also undergoes gene conversion, which plays a key function in regulating SabA levels. Adhesion is affected by emerging subpopulations of *H. pylori* with variable expression of the protein, which is a consequence of having recombination amongst *sab A*, *sab B* and *omp27* genes [107]. SabA also contributes significantly to spasmolytic polypeptide-expressing metaplasia (SPEM), which succeeds chronic atrophy and is a strategy for the stomach to reform its normal structural units following injury. It is thought that *H. pylori* can help SPEM progression, in which SabA adhesion to sLe^x^ plays a pivotal role [108].

**Hop P and gastric disorders:** Numerous studies have investigated an association between SabA and clinical outcomes. It appears that SabA is responsible for inflammation and its presence is correlated with clinical outcomes [109,110]. Also, a close relationship between this protein and gastric cancer has been found. In one study, 66% of *H. pylori* strains in patients with gastritis were SabA-positive, 44% were positive in individuals with duodenal ulcers and 70% in cases of gastric cancer [111]. Other studies that examined the frequency of SabA reported 93%, 86%, 80% and 23% detection in *H. pylori* strains in the Netherlands, France, Taiwan and Iran, respectively [58,112,113,114]. Recently, a Brazilian report revealed that SabA can accelerate gastric cancer in infected people [115].

#### 2.2.4. Hop Q

Otherwise known as Omp27, HopQ is classified into two families, HopQI and HopQII [36,116]. Both 3′ and 5′ ends of *hopQ* alleles are highly conserved in *H. pylori*, but divergence occurs in the 1.1 kb mid-region, with a 75–80% similarity of nucleotide sequence. However, they are different in terms of geographical distribution, HopQI being isolated mostly in East Asia and HopQII commonly present in western countries [117,118]. Similar to other OMPs, these proteins mediate adherence to the gastric mucosa. It seems that there is a correlation between HopQ and other virulence factors like CagA and VacA [119]. Prevalence of this protein is common in those *H. pylori* strains with *cag* PAI, which is responsible for encoding CagA and a type IV secretion system [47].

A family of receptors defined as carcinoembryonic antigen-related cell adhesion molecules (CEACAMs) is recognized for HopQ and HopQ. CEACAM activation interferes with immune functions of T and NK cells [120,121]. Moreover, CEACAMs mediates various cell functions such as adhesion, proliferation, immune response and motility. CEACAM1, 5 and 6 are expressed by gastric epithelial cells. CEACAM1, 3 and 4 have both cytoplasmic and transmembrane domains, while CEACAM5, 6, 7 and 8 have glycosylphosphatidylinositol linkage to the host cell membrane. A strong connection between HopQ and CEACAM1, 3, 5 and 6 N-terminal domains facilitates *H. pylori* adhesion to gastric epithelial cells. Interestingly, CEACAM1, 5 and 6 are found in multiple organs. Binding between HopQ and CEACAMs plays a crucial role in CagA delivery into host cells [121,122,123].

The relationship between HopQ and CagA modulation is a focus of research interest [124]. It has been shown that inflammatory reactions follow T4SS activation and transfer of CagA oncoprotein via HopQ-CEACAMs interaction. The inflammatory response ultimately leads to gastric cancer, which supports the idea of therapeutic approaches targeting HopQ-CEACAMs [125,126]. This interaction affects human CEACAMs, responsible for CagA activation and phosphorylation in polymorphonuclear neutrophils (PMNs) but not dendritic cells and macrophages. In PMNs it lessens CagA translocation and alters expression of CEACAM receptors. Also, the presence of human CEACAMs on PMNs increases bacterial survival within phagosome, thus resisting phagocytosis [127].

**Hop Q and clinical disorders:** The correlation between both types of HopQ and gastric cancer is established [128,129]. Also, a high incidence of gastric cancer has been reported in patients with *hopQI* and *vacA s1m1*, or with *hopQII* and *vacA S2* genotypes [130]. In two studies, in specific geographical regions in Iran, the rate of *hopQII* was higher than that of *hopQI* and a correlation between these OMPs and clinical outcomes was observed. However, another study showed the inverse result by which HopQI prevalence was higher with no association with gastrointestinal disorders [131,132]. Although its correlation with gastric diseases was demonstrated in several investigations, paradoxically HopQ could even be used therapeutically, as trials have shown good efficacies against melanoma metastasis [133].

#### 2.2.5. Hop S, Hop T and Hop U

HopS, HopT and HopU were first identified as blood group antigen-binding adhesin A (BabA) or OMP 28 (~80 kDa), BabB or OMP 19, and BabC or OMP 9, respectively. They each mediate attachment of *H. pylori* to histo-blood group antigens on gastric epithelial cells except for BabC, the function of which is not yet clear. Notably, there is extensive homology at the 3′ and 5′ segments of *babA* and *babB* [134]. There are two types of *babA*, namely *babA1* and *babA2*, with the latter divided into two subtypes with high and low protein production (Bab A-H and Bab A-L) [135,136]. An evaluation of glycosphingolipids as a receptor reported that *H. pylori* varies in its attachment to different blood groups including A Rh^+/−^ and O Rh^−^. Moreover, *H. pylori* could not adhere to glycosphingolipids in people with blood group O but could bind extremely well in A Rh^+/−^ individuals. In this study, Le^b^ hexaosyceramide, pentaosylceramide, heptaosylceramide, lactosylceramide, lactotetraosylceramide, neolactohexaosylceramide and pentaosylceramide were reported as BabA receptors [137]. In addition to Le^b^, fucosylated blood group A, B and O antigens are noteworthy receptors [138]. Depending on the mid region and ability to bind ABO antigens, there are two classifications of BabA, specialist and generalist. The former refers to those *H. pylori* strains that can attach to ALe^b^ (A-Lewis a), whereas the latter refers to those that bind to ALe^b^, BLe^b^ (B-Lewis b) and Le^b^ [139]. Also, analysis of variation in *babA* and *babB* revealed that there are five and three groups of alleles, including AD1-5 and BD1-3, for BabA and BabB, respectively [136].

*Helicobacter pylori* is able to achieve compatibility with the variable gastric acidic environment through recombination and mutation in *babA* genes. This enables mediated attachment via this protein, which is responsible for this phenomenon, thereby increasing the risk of progression to gastric cancer [140]. BabA is an antigen that is commonly expressed by *H. pylori* and which is related to specific clinical outcomes including peptic ulcers and gastric cancer. Also, colonization occurs predominantly in the lowermost antrum of the stomach [141,142]. Based on recent studies, recombination between the three *bab* genes frequently happens [143]. BabA undergoes genetic regulation through phase variation, which modulates its role in adherence. Also, it can be affected by recombination between *babA* and *babB* genes [144]. This genetic regulation is beneficial for *H. pylori* adaptation to its gastric environment in which the bacterium is exposed to a high level of physiological stress [145].

**Correlation with gastric disorders:** Several studies have investigated a correlation between *babA* gene expression and gastric disorders such as peptic ulcers and gastric cancer. Reportedly, inflammation induced by BabA adhesion results in gastric conditions such as precancerous transformations and intestinal metaplasia [146,147,148]. Also, a correlation between *Le^b^* and low binding activity and risk of duodenal ulcers was found [149]. Notably, undertaken the correlation between this genotype and gastric cancer was demonstrated separately in Germany, Portugal, Japan, Taiwan, China, USA and Brazil [91,150,151,152,153,154]. Similarly, *babA2* gene was recently found at high frequency in patients with gastric cancer or peptic ulcers, although discrepancies arise regarding whether or not development to the severe gastric condition is associated with this genotype. A possible reason for this could be a lack of expression of BabA protein despite the presence of the gene [67]. This agrees with a meta-analysis of twenty studies that indicated a strong association between BabA2 and increased risk of gastric cancer in Asian populations compared to South American ones, suggesting a significant role of this virulence factor in pathogenicity [155].

#### 2.2.6. Hop Z

This protein, also known as HP9, has a role in adherence to gastric epithelial cells, although its receptor is not yet recognized [72]. The *hopZ* gene undergoes slipped-strand mispairing and is regulated by a phase-variable CT repeat, which means whether it is switched on or off depends on the prevailing in vivo situation. There are two types, HopZI and HopZII. This differentiation dates to the era in prehistory before migration of humans from Africa [69,156]. Its relationship with infection is suggested by some findings [157,158]. In one investigation, an association between this protein and gastric cancer was reported, but a correlation between HopZ and chronic atrophic gastritis has yet to be found [72,156].

#### 2.2.7. Hop V, Hop W and other OMPs

These porin members belong to the Hop A/E family. This is due to homologous N-terminal and C-terminus regions. In terms of their pore size, HopV and HopW are similar to *E. coli* OmpF porin. Among *H. pylori* isolates, their expression is relatively less. Hop X/Y have been identified as porins similar to Hop A-D [37,159,160,161]. Colonization attributed to OMPs is mediated by *H. pylori* OMP 18 [162].

In the *Helicobacter* outer membrane (Hom) family, four members (HomA, B, C and D) are recognized, of which HomB is the most studied. Hom A/B exhibit variation in regard to genes copies and genomic localization in different geographical areas. The rate of homology between *homA*/*B* genomes is estimated at 90%, with only a 300 bp difference. Similar to other OMPs, recombination and phase variation are involved in gene duplication [37]. HomA/B are known for their significant roles in adherence, antibiotic resistance, biofilm formation and gastric malignancies [163]. Two important functions ascribed to HomB are IL-8 secretion and adherence [164]. While no specific association with clinical outcomes has been found for either of these proteins, they are likely to be involved in *H. pylori* persistence [165].

Another group, defined as Hof proteins, includes eight members, namely Hof A-H. Each of these contains a hydrophobic C-terminal motif, similar to the Hom family. Recently, a study of *Helicobacter heilmannii* showed that Hof E and Hof F act as adhesins in the same way as other OMPs [37,166].

## 3. Advancements in *H. pylori* Vaccine Development and Therapeutic Strategies

### 3.1. Targeting Outer Membrane Proteins

With the rise of antibiotic-resistant strains of *H. pylori*, there is a growing emphasis on exploring alternative treatments to antibiotics. Consequently, the development of an *H. pylori* vaccine has emerged as a prominent and actively researched area (Table 2).

Two types of vaccine, whole-cell bacterium and a recombinant preparation, which combines protective antigens with immune adjuvants, are considered the main approaches [183]. While development of the former was abandoned for various reasons, including complexity of vaccine production, the latter has progressed. Different immune adjuvants, including BabA, SabA, OipA, CagA, and VacA, have provided vaccines with higher protective effects [184].

Targeting OMPs is a promising innovative therapeutic approach against *H. pylori*, given their importance and roles in gastric conditions (Table 3). Hop B and Hop C are considered potential targets for vaccine therapy as they are involved during the early colonization stages of infection with *H. pylori.* In immunization studies in mice, when HopB, either on its own or in combination with other antigens (BabB, urease, catalase), was conjugated to the DC-Chol mucosal adjuvant, enhanced cellular and humoral protective responses were observed [185]. In a further promising evaluation of HopB immunogenicity in recombinant plasmids, HopB recombinant protein was introduced as a novel means of infection prophylaxis and eradication [186].

It has been shown that OipA is a promising candidate for an oral vaccine. In murine studies, inoculating IgA raised against OipA significantly ameliorated *H. pylori* infection [187]. Similarly, a *Salmonella typhimurium* bacterial ghost-based DNA vaccine that delivers the *oipA* gene is proposed as a novel immunogen. This oral vaccine was capable of boosting immune responses, observed as heightened antibody and cytokine levels, and minimized bacterial colonization [188]. In another study, the efficiency of a recombinant OipA vaccine in mice was indicated by an elevated interferon-γ response [189]. Hence, a series of investigations shows a connection between this protein and gastric disorders, but using a combination of factors is suggested as a more accurate predictor of clinical outcomes.

**Table 3 pathogens-13-00392-t003:** Different *H. pylori* OMPs, their given names, receptors, and roles in gastric disorders.

OMP	Also Known as	Receptor	PU	GC	GA	DU	MALT	Reference(s)
Cag A	Cytotoxicity in associated gene	Epithelial cell	✓	✓	EPIYAD/C	-	✓	[54]
					Cag L			
Vac A	Vacuolating cytotoxin	RPTP-α	VacAs1m1 VacAs2m2	VacAs1m1	VacAs1m1	-	-	[43,54]
		RPTP-β						
		Lipids						
		Heparin sulphate						
		Sphingomyelin						
		Fibronectin						
		Β2-integrin						
		EGFR						
Hop B/C	Alp A/B	Laminin	-	-	-	-	-	[190,191]
		Collagen IV						
Hop H	Oip A	Not known	✓	✓	-	✓		[54,88,91]
Hop P	Sab A	sLe^x^	-	-	-	-	✓	[58]
		sLe^a^						
Hop S	Bab A	sLe^b^	Bab A2	✓	✓	-	-	[54]
		A, B, O blood group						
Hop Q	-	CEACAMs	-	✓	-	-	-	[121]

Abbreviations: PU, peptic ulcer; GC, gastric cancer; GA, gastric adenocarcinoma; DU, duodenal ulcer; MALT, mucosa-associated lymphoid tissue; sLe, sialyl-Lewis; EGFR, epidermal growth factor receptor; ✓, genopositive.

Porcine milk has shown promising therapeutic potential by interfering with SabA adhesion. Apparently, this product has an inhibitory effect on *H. pylori* adhesion by expressing Lewis B glycans, as well as sialyl Lewis X [192]. Multiple trials demonstrated the efficacy of SabA as a potential recombinant vaccine candidate. One study evaluated a novel immunogenic cocktail, including VacA, BabA, SabA, FecA and Omp16, using a reverse vaccinology approach [193,194]. Similarly, a multi-epitope oral vaccine composed of BabA, SabA, OipA, VacA, CagA, cholera toxin subunit B (CTB) and other components, serves as another promising vaccine candidate [195]. The significant role of SabA in anchoring *H. pylori* and its ability to adapt to the gastric environment reinforces the idea of using this protein as the basis for vaccine design. Furthermore, there is promise in evaluating glycosphingolipids as therapeutic targets to develop new treatments for pathogenic host–microbe interactions in the human stomach [196,197].

Research has been performed to evaluate anti-adhesive agents on *hopQ* genotypes. In one study, HopQ1 was more sensitive than HopQII to different dietary ingredients [198]. It is suggested that engineered CEACAMs conjugated to antimicrobial agents, with higher specificity and affinity for HopQ, can improve antibacterial efficiencies [199]. Further research is needed regarding prevalence of this protein, correlation with clinical outcomes and potential targets.

When the efficacy of various drugs to interfere with the interaction between BabA and gastric mucosa was evaluated, rhamnogalacturonans showed potential as inhibitors of this protein [200]. In terms of treatment, some progress has been made using mucolytic agents. Findings show that N-acetylcysteine has the ability to disrupts BabA adhesion to the gastric mucosa. Also, this conserved disulfide has a synergic effect with antibiotic therapy that boosts the efficiency of each [201]. Pectin and rhamnogalacturonans show promise as BabA adhesion blockers, indicating that BabA could potentially serve as a target for designing receptor-mediated adhesion drugs [200].

Four highly conserved OMPs have been discovered that offer considerable potential as vaccine candidates. These proteins, namely HopV, HopW, HopX, and HopY, show no signs of phase variation, indicating their stable expression during chronic infection and suitability as immunogens [161,184].

### 3.2. Targeting Cag A

Regarding therapies that target Cag A, progress has been made using ATPase Cagα inhibitors to target the Cag type IV secretion system. CHIR-1, a kinase-targeting compound, and difluoromethylornithine, have both shown promising results. However, technical limitations make it difficult to achieve full inhibition. It is necessary to preincubate bacteria with CHIR-1 to reach the strongest level of inhibition [202,203,204,205]. In summary, a major role of VacA and CagA in *H. pylori* pathogenicity and disease progression is evident. Irrespective of the geographical differences, CagA is a good indicator of patient outcome and targeting this protein could provide a potentially effective means of treatment [206].

### 3.3. Enhancing Immune Responses

In order to achieve a protective immune response some therapeutic strategies such as T cell activation and targeting inhibitory receptors are of note. In this regard, promising results were obtained using MDX-1106 (anti-PD-1), lambrolizumab, and rapamycin to control the mTOR/p70 S6 kinase pathway. However, further clinical research is required to acquire a deep knowledge of immunity to *H. pylori*. A better understanding of these mechanisms is critical to design an appropriate vaccine [207]. Several prototype vaccines are in development and currently undergoing trials (Table 2).

Regarding other therapeutic approaches directly against *H. pylori*, several targets are suggested for treatment. These include shikimate pathways (involved in ubiquinone and aromatic acid synthesis), flavodoxin (electron carrier protein), coenzyme A, succinylase pathway, and urease inhibitory compounds. By developing compounds that interfere with these targets, researchers aim to disrupt essential bacterial functions and reduce colonization by *H. pylori*, leading to its control or even eradication from within the host [208]. As discussed below, there are several innovative therapeutic approaches against *H. pylori*, including novel treatments targeting key virulence factors and host–microbe interactions.

## 4. Challenges to *Helicobacter pylori* Vaccine Development

### 4.1. Genetic Characteristics of H. pylori OMPs Contribute to Its Variability

A feature of the genome of *H. pylori* is its appreciable plasticity. This is due to genetic recombination which results in a high level of mutation, notably reported for *babA2* gene expression [209,210]. This pathogen uses various micro- and macro-diverse tools to survive in the gastric mucosal environment [211,212]. Genetic incongruity is especially pronounced among *omp* genes [37]. Most studies have been performed on two *H. pylori* well-researched strains, 26,695 and J99, which are thought to be representative of clinically significant isolates [116].

There are three categories of *H. pylori* genes: phase-variable; structure-variable; and strain-specific [26]. Some phase-variable genes use a specific method to escape from immune surveillance whereby not only does the expression of antigens change, but also the bacterium becomes more heterogenous. To date, six genes, including *sabA/B*, *babB/C*, *oipA* and *hopZ*, have been identified that are regulated by this mechanism [26,213].

One interesting finding is that *H. pylori* can upregulate expression of *Le^b^* and *Le^x^*, yielding BabA and SabA receptors, respectively. This function is performed by deposition of these antigens, which facilitates increased colonization [214].

There is broad similarity between *H. pylori* strains in terms of ribosome-binding sites (nucleotide number). However, the shorter spacing that is observed in some *H. pylori* genes may cause a change in the gene expression reported for seven orthologous pairs of *omp* genes. Examples can be seen in *babA* genes. Slipped-strand repair plays a pivotal role in altering expression of these proteins, thereby providing a mechanism by which *H. pylori* can evade the host immune system. The Com-B system in *bab A*/*B*/*C* is integral to this mechanism. While the central region of these genes is diverse, the 5′ and 3′ ends are similar. Slipped-strand repair has been reported in several genes and is thought to underpin antigenic variation and genetic diversity that is observed among *H. pylori* strains. Five *hop* orthologs undergo this regulation to change signal sequence, while the final product of expression remains the same [36,37,136,210,215]. In addition, gene duplication, in which there are two copies of an allele, is described for *babA*, *hopJ*, *hopK*, *hopQ*, *hopM* and *hopN* genes. This event differs between various *hop* genes depending on the *H. pylori* strain [37,134].

As well as OMPs, based on recent studies there are two other genes that affect *H. pylori* pathogenicity. *ice A1* and *ice A2* are a pair of novel genes that are considered as risk factors for various gastric conditions. Transcription of either *ice A* gene can be induced by *H. pylori* attachment to gastric epithelial cells. Their distribution among different geographical areas and gastric diseases is variable [216,217]. There is a relationship between this gene and other virulence factors such as CagA and VacA. The findings of one study suggest that *ice A* and *cagA* may be used as potential markers for clinical outcomes [62]. However, as the findings are paradoxical, elucidation through further research is required [59,218]. The duodenal ulcer-promoting gene *dupA*, which is located in the plasticity region of *cag* PAI, is thought to provide an increased risk for duodenal ulcers. Expression of this gene induces IL-8 and neutrophil activity [219]. On the other hand, in patients with gastric cancer its prevalence is much lower. *dup A* may provide a good candidate to predict clinical outcomes such as duodenal ulcers [26,219,220,221].

### 4.2. Protective Nature and Heterogeneity of Biofilm Limit Vaccine Accessibility

Another medical challenge that is presented by *H. pylori* is its capacity to form biofilm. Under the protection of the impervious matrix of extracellular polymeric substances (EPS), bacteria are refractory to antibiotic penetration, thus greatly reducing the efficacy of standard treatment approaches [222]. There is strong evidence for a direct correlation between biofilm formation and antibiotic resistance, influenced by factors such as OMPs, other virulence factors, extracellular matrix, efflux pumps and metabolic changes [223]. Therefore, susceptibility to antibiotics such as amoxicillin, clarithromycin, levofloxacin, and metronidazole by bacteria protected by biofilm is reduced substantially [224].

Heterogeneity in the regulation of OMPs, an important feature of biofilms, leads to variation in biofilm composition and plays a key role in adherence [225,226]. Of all OMPs in *H. pylori*, the Hom family and AlpB are initially involved. Since the latter is highly conserved in *H. pylori* strains, it is considered a promising therapeutic candidate [227,228].

Another potential target for vaccine design has been found in *H. pylori* biofilm. OMVs are small spherical structures released by Gram-negative bacteria. They are an integral component of *H. pylori* biofilm EPS matrix. Produced during bacterial growth, OMVs are implicated in pathogenesis through biofilm formation. A recent study demonstrated that an α-class carbonic anhydrase (CA) is found in OMVs, synthesized by both biofilm-producing and planktonic *H. pylori* strains [229]. On the other hand, genes responsible for encoding CAs in *H. pylori* are distributed to cytoplasmic β-CA (*hpβCA*) and α-CA (*hpαCA*). Also, expression of these two genes is accelerated at low pH and their joint activity with urease helps *H. pylori* to withstand the acidic gastric conditions. Therefore, hpCA has been considered as a new therapeutic candidate [229,230,231,232,233].

### 4.3. Overcoming Immune Tolerance of H. pylori

The protective host immune response to *H. pylori* helps to lessen the threat that colonization poses. However, this noted pathogen has evolved a unique strategy to overcome host defenses. Long-term infection is a consequence of remodeling of the host-pathogen interface as well as immune evasion due to expression of multiple virulence factors [234]. Hence, *H. pylori* poses a challenge to therapeutic approaches and effective vaccine design by modulating host immunity and inducing immune tolerance. Achieving sufficient and durable protection that involves eliciting robust and long-lasting immune responses is warranted.

Different *H. pylori* virulence factors cause immune tolerance through various ways. It is known that *cag* PAI is a potent driver of IL-8 and NfκB secretion. A cascade of intracellular activities is involved in *H. pylori*-dependent signal transduction. Nucleotide-binding oligomerization domain (NOD)1 is an intracellular pattern recognition receptor that recognizes bacterial peptidoglycan, among other danger signals, and thus plays a fundamental role in innate and adaptive defenses and control of inflammation1 [235]. This protein also has an important function in cancer development. Following interaction between CagA and PAR1b, BRCA1 disturbs phosphorylation, which can lead to the promotion of DNA double-strand breaks and BRCAness. This phenomenon is expanded via p53 inactivation, enabling DNA-damaged cells to escape from apoptosis and proliferate. This propels a “hit-and-run mechanism”, which is a significant cause of gastric carcinogenesis [236].

Similarly, VacA activity affects the immune system in multiple ways. For example, paracellular permeability during carcinogenesis, TGF-β1 production and heightened inflammation. At the same time, through vacuolization, antigen proteolysis decreases, which subsequently reduces peptide presentation and thereby inhibits T cell stimulation. This leads to down-modulation of CD4^+^ and CD8^+^ responses, thus facilitating the persistence of *H. pylori* infection [237].

Composed of lipid A and polysaccharide, LPS is an *H. pylori* endotoxin within the outer cell membrane. It is intrinsically involved in septic shock and sepsis through triggering proinflammatory mediators such as TNF-α and IL-1 [238]. LPS can promote gastric cancer through inhibiting inflammatory immune responses as well as preventing invasion of gastric cancer cells by immune mediators. Specifically, LPS derived from *H. pylori* can weaken cytotoxicity of mononuclear cells towards gastric cancer cells, as well as cytotoxic activity of gastric epithelia and NK cells. In addition, *H. pylori* LPS selectively elevates production of IL-18 and IL-12 and activates signal transduction patterns related to TLR4- and toll/IL-1 receptor [238,239,240]. Multiple studies have shown the significant role of TLR4-LPS in initiation and escalation of gastric cancer. The synthesis of two important factors in promoting cancer, TNF-α and IL-8, is accelerated after TLR4-LPS binding [241,242].

A series of mechanisms is utilized by this pathogen to enable it to evade the host immune system. *H. pylori* is a motile bacterium that is equipped with at least four flagella, the coordinated actions of which propel it through the gastric mucus layer. Each flagellum comprises several components including hook, basal body, filament and sheath. Interestingly, reports show that they contribute to biofilm formation [237,243]. As alluded to above, urease production is another means to combat the immune system. As it helps to alleviate the acidic environment, this implies a role in chronicity of infection and bacterial persistence. Also, ammonium produced by urease may cause damage to host cells [244,245]. Additional recognized roles for urease include a chemotactic effect on immune cells as well as angiogenesis, which may promote development of infection to gastric cancer [246,247].

A “founder colony” is a newly proposed model of escape and persistence in which *H. pylori* penetrates deep within the microenvironment of gastric glands to initiate colonization. These small colonies then expand to form persistent clonal population islands. They are distinct from planktonic bacteria in the superficial layer and gaining the space for new bacterial growth presents a challenge [248].

### 4.4. The Wait for an Approved Vaccine

Despite almost 40 years of research and development, no *H. pylori* vaccine is commercially available, with most clinical trials concluding after phase I. In addition to genetic diversity, biofilm characteristics, and the risk of exacerbating gastric diseases and autoimmunity due to an aberrant immune response, other reasons may partly account for this. For example, intracellular features of *H. pylori* enable it to effectively ‘hide’ inside gastric epithelial cells and gastric lamina propria, thus contributing to persistent infection [249]. Another concern is that many preclinical studies have been performed in mice, which are not natural hosts of *H. pylori*. Hence, any vaccine efficacy observed in mouse models may not translate accurately to humans [250]. Enhancing investment and prioritizing research into the design of an efficacious *H. pylori* vaccine are public health imperatives considering the widespread prevalence and significant disease burden associated with this bacterium.

## 5. Conclusions

*Helicobacter pylori* is the principal cause of gastric conditions including peptic ulcers, gastric carcinoma, mucosa-associated lymphoid tissue lymphoma and gastritis. Its pathogenicity is due to a combination of virulence factors including urease, flagella, and OMPs [22,191,251,252,253,254]. The progression of *H. pylori* infection to gastric cancer happens through a series of events. Primary inflammation may develop into acute gastritis and chronic gastritis. At this stage, multiple factors such as stomach pH, genetic diversity and environmental factors can gradually alter the gastric condition to cancer. During the early stages, most patients are unaware of their condition and so treatment is not started until symptoms are more advanced. Hence, developing earlier and more accurate screening methods to enable prevention and eradication of *H. pylori* at the community level, as well as better treatment strategies to combat existing infection in patients, are warranted [255]. This will require an in-depth knowledge of different features of this bacterium.

Various indices are involved in determining clinical outcome. Notable among these are host genetics, particularly relating to the functioning of an individual’s immune system, as well as *H. pylori* virulence factors These belong to one of three groups that relate to colonization, development, and disease [90,256]. OMPs exist in all *H. pylori* strains as a tool for initial attachment, so are therefore considered as potential targets for candidate vaccine design. Recombinant vaccines incorporating CagA, VacA, urease, BabA, SabA, OipA and porin proteins show promise in ongoing trials [194].

Adherence to and colonization of epithelial cells play integral, initial roles in the pathogenicity of *H. pylori*. These interactions are mediated via OMPs and other virulence factors. Through adhesion to gastric mucosa and harnessing its type IV secretion system, the pathogen transfers toxins and effector molecules into the host cell. OMPs also facilitate inflammation, metaplasia and the ultimate pathological outcome of gastric cancer [257]. Each OMP has a distinct receptor, so gaining a clear understanding of them all aids diagnosis of infection and benefits clinical outcomes. In this context, combined evaluation of different OMPs can be both more rapid and accurate than any single identification. In one study, by considering various OMPs including OipA, BabA and SabA, the accuracy of gastric cancer prediction reached 77% [258]. In addition, an association between the production of some OMPs has been identified. While there is an inverse relationship between OipA and CagPAI, the presence of this OMP is a prerequisite for CagA translocation [86]. Similarly, at an 83% rate of *H. pylori* infection, a close relationship between VacA and chronic gastritis is apparent. Also, a correlation between VacA, BabA2 and OipA with increased risk of gastric cancer has been revealed [103]. When considering these relationships as potential prognostic markers a number of challenges such as limited time of survival and geographical regionality of occurrence should be considered.

Although several *H. pylori*-related virulence factors are involved in promoting gastric disorders, the causal relationships that underlie severe gastric conditions are still to be elucidated. The importance of these factors is crucial from both treatment and management perspectives [71]. It is also necessary to gain a precise evaluation of the epidemiology of each OMP, as its prevalence at a population level is different based on geographical region, even within the same country. Other criteria such as patient age gender, and bacterial genotype are also important [259,260,261]. To date, despite considerable research efforts there is no vaccine candidate that is sufficiently far advanced to be of interest to a pharmaceutical company to take through commercial development. Therefore, further research and greater investment are warranted in order to improve vaccine design and efficacy in terms of both prevention and lessening medical burden.

## Figures and Tables

**Figure 1 pathogens-13-00392-f001:**
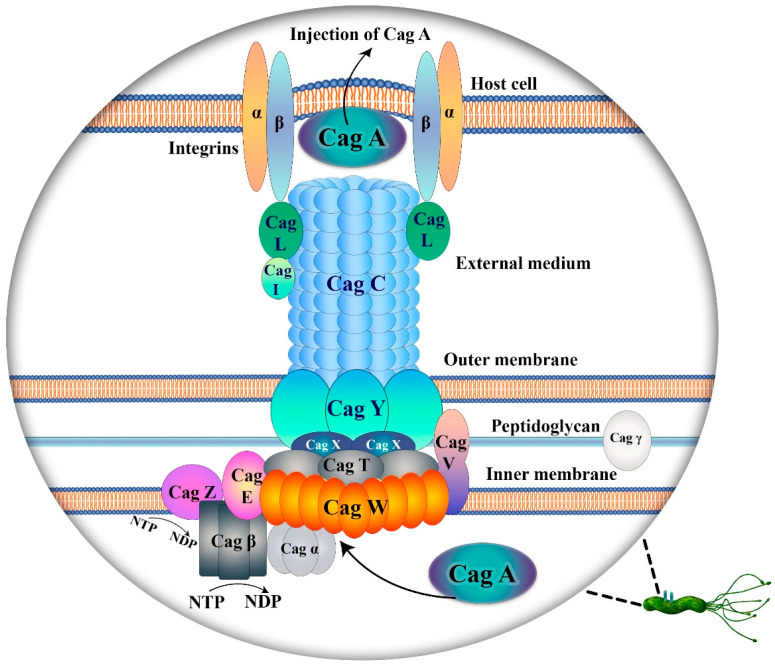
*Helicobacter pylori* type IV secretion system (T4SS) and Cag A pathogenicity. In the intricate interplay between *H. pylori* adhesins and epithelial cells, various receptors play a crucial role in mediating binding. A noteworthy homology has been observed between *H. pylori* outer membrane proteins and Vir proteins in *Agrobacterium*., The *cagA* pathogenicity island consists of distinct elements within the multicomponent T4SS complex. Specifically, Cag X, T, and Y contribute to the core complex, while CagE, W, and V participate in the inner membrane complex. Additionally, CagC, L, and I are instrumental in pilus formation. Subsequent to the interaction between host cells and binding proteins, the CagA substrate is delivered through assembled pili. Integrin receptors play a pivotal role in facilitating this interaction with CagA, Y, L, and I. In the lower section of the diagram, CagE, Z, α, and β are implicated in generating energy through dephosphorylation of nucleoside triphosphates (NTP), ultimately leading to translocation of CagA. Notably, Cagγ, situated in the peptidoglycan layer, assumes responsibility for hydrolyzing peptidoglycan.

**Table 1 pathogens-13-00392-t001:** Classification of *Helicobacter pylori* outer membrane proteins.

Protein Family	Number of Sub-Family	Sub-Family Genes
Hop	22	*hopZ*, *hopD*, *hopM*, *hopA*, *hopF*, *hopG*, *hopJ*, *hopH*, *hopE*, *hopO*, *hopP*, *hopC*, *hopB*, *hopK*, *hopI*, *hopL*, *hopQ*, *hopN*, *hopU* *babA*, *babB*
Hor	12	*horA*, *horB*, *horC*, *horD*, *horE*, *horF*, *horG*, *horH*, *horI*, *horJ*, *horK*, *horL*
Hof	8	*hofA*, *hofB*, *hofC*, *hofD*, *hofE*, *hofF*, *hofG*, *hofH*
Hom	4	*homA*, *homB*, *homC*, *homD*
FecA-like	3	*fecA-1*, *fecA-2*, *fecA-3*
FrpB-like	3	*frpB-1*, *frpB-2*, *frpB-3*
Efflux pump	6	*hefA*, *hefD*, *hefG* *flgH* *palA* *lpp20*

**Table 2 pathogens-13-00392-t002:** Improvement of vaccine design and progression of development pathway.

Vaccine	Type	Status	Reference	Time
Urease	Oral Recombinant	I	[167,168,169,170,171,172,173,174]	1996–2004
Whole cell	Oral	I	[175]	2001
Imevax/IMX101	Multicomponent	I	[176]	Ongoing
VacA, CagA, NAP (NCT00736476)	Recombinant	I/II	[177]	2018
HpaA expression by *Vibrio cholera*	Recombinant	Preclinical	[178]	2017
Cholera toxin B and *H. pylori* Lpp20	Epitope	Preclinical	[179]	2016
*H. pylori* vaccine	Oral recombinant	III	[180]	2015
CTB-UreI-UreB (BIB)	Recombinant multi-epitope	Preclinical	[181]	2014
HelicoVax	Multi-epitope	Preclinical	[182]	2011

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
