# Peer review of "Helicobacter pylori Outer Membrane Proteins and Virulence Factors: Potential Targets for Novel Therapies and Vaccines"

_pathogens, 2024, doi:10.3390/pathogens13050392_

Round 1

Reviewer 1 Report

Comments and Suggestions for Authors

To the authors,

It is a well-reviewed paper about the outer membrane protein of helicobacter pylori and it may be very helpful for doctors or scientists who want to know about the pathogenesis of stomach disease-related helicobacter pylori.

Minor revision:

1. as you mention in the article OMPs are good target for vaccine for helicobacter pylori and you insist may the vaccine will be developed which target OMPs. but the 40 years after discover helicobacter there is no vaccine for commercial which summit to use to patients.

please explain your opinion on why there is no vaccine developed and what is the clue to solve this problem in your review.

thanks

Author Response

Reviewer 1

In-text changes shown in green in marked-up copy of revision.

Comments and Suggestions for Authors

To the authors,

It is a well-reviewed paper about the outer membrane protein of Helicobacter pylori and it may be very helpful for doctors or scientists who want to know about the pathogenesis of stomach disease-related Helicobacter pylori.

Minor revision:

  1. as you mention in the article OMPs are good target for vaccine for Helicobacter pylori and you insist may the vaccine will be developed which target OMPs. but the 40 years after discover Helicobacter there is no vaccine for commercial which summit to use to patients.

Please explain your opinion on why there is no vaccine developed and what is the clue to solve this problem in your review.

? This issue is now addressed directly in a summary statement (page 13, bottom paragraph; section 3.4). Before this, the unique challenges to vaccine development that H. pylori presents are explained in detail in sections 2 and 3, with expanded sections 2.1, 3.2 and 3.3 (pages 9-10, 12) (also accommodating the suggestion of Reviewer 2).

Please see the attachment for further information.

Reviewer 2 Report

Comments and Suggestions for Authors

The content covered in this review is interesting, but there are still numerous logical and writing issues throughout the text. The main problems include:

1.     The topic of "Treatment Strategies" in the title does not belong to the "Complexity of Helicobacter pylori Pathogenicity."

2.     The overall structure of the text is unclear, with each section using the same numbering, making it difficult to read and understand.

3.     There are many detailed errors in the article, such as "HopQ" appearing twice consecutively in Table 1, It is recommended to carefully revise the entire text.

4.     For the section related to target selection for vaccine development, it is suggested to carefully organize it based on the required characteristics of Helicobacter pylori vaccines.

Author Response

Reviewer 2

In-text changes shown in red in marked-up copy of revision.

Comments and Suggestions for Authors

The content covered in this review is interesting, but there are still numerous logical and writing issues throughout the text. The main problems include:

  1. The topic of "Treatment Strategies" in the title does not belong to the "Complexity of Helicobacter pylori Pathogenicity."

? The title is now revised to: Helicobacter pylori Outer Membrane Proteins and Virulence Factors: Potential Targets for Novel Therapies and Vaccines. This now better fits the content of the review (also accommodating the suggestion of Reviewer 4).

  1. The overall structure of the text is unclear, with each section using the same numbering, making it difficult to read and understand.

? The structure of the paper follows very closely the journal guidelines for sections and subsections. Accordingly, the numbering is sequential and hierarchical.

  1. There are many detailed errors in the article, such as "HopQ" appearing twice consecutively in Table 1, It is recommended to carefully revise the entire text.

? We thank the reviewer for pointing out this duplication, which is now corrected. Furthermore, we have checked the revised paper for any other minor typographical oversights and inconsistencies of presentation.

  1. For the section related to target selection for vaccine development, it is suggested to carefully organize it based on the required characteristics of Helicobacter pylori vaccines.

? Section 2, to which the Reviewer refers, is restructured accordingly with revised or new subsection titles (pages 9-11). The text is expanded to now describe more clearly vaccine development in terms of how to target the pathogen (sections 2.1, 2.2, 2.3) (also accommodating the suggestion of Reviewer 1).

Please see the attachment for further information.

Reviewer 3 Report

Comments and Suggestions for Authors

Very well written review, but it will be much better if the authors can provide a figure that summarizes interaction of all virulence factors mentioned in this review.

Comments on the Quality of English Language

English language is fine

Author Response

Reviewer 3

Comments and Suggestions for Authors

  1. Very well written review, but it will be much better if the authors can provide a figure that summarizes interaction of all virulence factors mentioned in this review.

–   We acknowledge the suggestion of the Reviewer seeks to provide an overview for the reader. However, as most virulence factors that are discussed are expressed and function independently of each other, it is not possible to show schematically any supposed interactions between them.

Please see the attachment for further information.

Reviewer 4 Report

Comments and Suggestions for Authors

There are several concern regarding the manuscript:

- I can confirmed that this manuscript was focused on the role of OMPs on bacterial pathogenesis. Therefore, the title should be modified to cover the main idea of the manuscript.

- The aim of this work was not highlighted in the introduction e.g., particularly last paragraph of the manuscript.

- H. pylori should stated with italic font through the full-text.

- What is your purpose from discuss about CagA and VacA ?? the manuscript was discussed mainly about OMPS, so these section was not essential. In addition, there is not new and interesting information has been suggested in CagA and VacA sections. The author should discuss regarding possible association between OMPs and CagA/VacA virulence factors.

- The manuscript was poorly organized, the results, introduction, conclusion should be separated with exclusive subsections.

- The author should cited relevant update meta-analyses regarding the association between OMPs and gastrointestinal diseases to consider more evidence with high confidence. 

- The authors should use table to summarized the animals studies and human studies regarding the impact of OMPs on digestive diseases and treatment outcomes.

- Discuss about relationship between OMPs and H. pylori antibiotic resistance as well as biofilm formation.

- In the section "Advancements and Challenges in Therapeutic Strategies Targeting H. pylori" the author should mainly discuss about the application of OMPs on H. pylori therapies. 

Author Response

Reviewer 4

In-text changes shown in purple in marked-up copy of revision.

Comments and Suggestions for Authors

There are several concern regarding the manuscript:

  1. I can confirmed that this manuscript was focused on the role of OMPs on bacterial pathogenesis. Therefore, the title should be modified to cover the main idea of the manuscript.

? The title is now revised to: Helicobacter pylori Outer Membrane Proteins and Virulence Factors: Potential Targets for Novel Therapies and Vaccines. This now better fits the content of the review (also accommodating the suggestion of Reviewer 2).

  1. The aim of this work was not highlighted in the introduction e.g., particularly last paragraph of the manuscript.

? A new paragraph to explain the paper’s aims now concludes the Introduction section (top of page 3).

  1. H. pylori should stated with italic font through the full-text.

? We thank the reviewer for pointing this out, which is now corrected. Furthermore, we have checked the revised paper for any other minor typographical oversights and inconsistencies of presentation.

  1. What is your purpose from discuss about CagA and VacA ?? the manuscript was discussed mainly about OMPs, so these section was not essential. In addition, there is not new and interesting information has been suggested in CagA and VacA sections. The author should discuss regarding possible association between OMPs and CagA/VacA virulence factors.

? CagA and VacA are pivotal virulence factors in H. pylori pathogenesis, influencing disease outcomes significantly. While the manuscript focuses primarily on OMPs, providing sections on CagA and VacA alongside enhances readers’ understanding of the range of virulence mechanisms. These discussions elucidate potential associations and synergies between OMPs, CagA, and VacA, which are crucial for unraveling the complexity of H. pylori infection. Moreover, recognizing the interplay between these virulence factors and OMPs is essential for designing effective therapeutic and vaccine strategies.

  1. The manuscript was poorly organized, the results, introduction, conclusion should be separated with exclusive subsections.

? While the organization of the paper has been revised to accommodate suggestions by other reviewers, the overall structure follows very closely the journal guidelines for sections and subsections. Accordingly, the numbering is sequential and hierarchical. As this is a review, there is no Results section.

  1. The author should cited relevant update meta-analyses regarding the association between OMPs and gastrointestinal diseases to consider more evidence with high confidence.

–   A comprehensive overview of these topics is outside of the remit of this article and would warrant a separate review, of which there are several recently published. We are also mindful of the already large number of references and the increase that a meta-analysis would require.

  1. The authors should use table to summarized the animals studies and human studies regarding the impact of OMPs on digestive diseases and treatment outcomes.

–   A comprehensive overview of these topics is outside of the remit of this article and would warrant a separate review, of which there are several recently published.

  1. Discuss about relationship between OMPs and H. pylori antibiotic resistance as well as biofilm formation.

? While the relationship between OMPs and antibiotic resistance in H. pylori, as well as biofilm formation, is indeed an important aspect of research, our aim is primarily to explore the roles of OMPs in H. pylori pathogenicity, vaccine development, and therapeutic approaches. However, where relevant, brief mentions of antibiotic resistance and biofilm formation are included (for example, see Section 3.2). A comprehensive overview of these topics is outside of the remit of this article and would warrant a separate review, of which there are several recently published.

  1. In the section "Advancements and Challenges in Therapeutic Strategies Targeting H. pylori" the author should mainly discuss about the application of OMPs on H. pylori therapies. 

? We consider this to be true, to which the Reviewer refers in their first point. In the revised text, this discussion is expanded, as exemplified by Section 2.1 (pages 9-10).

Please see the attachment for further information.

Reviewer 5 Report

Comments and Suggestions for Authors

This review is to provide a current overview of different H. pylori OMPs and discuss their pathogenicity, epidemiology and correlation with various gastric diseases.

It is well documented and summarizes leading and recent studies in this field.

It is well written.

But there is a slight confusion in the structure level.

For example:

1.1. Cag A and Vac A

1.1. H. pylori Outer Membrane Proteins

1.1.1. Hop B and Hop C

1.1.1. Hop H, a phase-variable protein

1.1.1. Hop P

1.1.1. Hop Q

1.1.1. Hop S, Hop T and Hop U

I recommend that it can be considered to be published in your journal after minor revision.

Author Response

Reviewer 5

Comments and Suggestions for Authors

This review is to provide a current overview of different H. pylori OMPs and discuss their pathogenicity, epidemiology and correlation with various gastric diseases.

It is well documented and summarizes leading and recent studies in this field.

It is well written.

  1. But there is a slight confusion in the structure level.

For example:

1.1. Cag A and Vac A

1.1. H. pylori Outer Membrane Proteins

1.1.1. Hop B and Hop C

1.1.1. Hop H, a phase-variable protein

1.1.1. Hop P

1.1.1. Hop Q

1.1.1. Hop S, Hop T and Hop U

I recommend that it can be considered to be published in your journal after minor revision.

? While the organization of the paper has been revised to accommodate suggestions by other reviewers, the overall structure follows very closely the journal guidelines for sections and subsections. Accordingly, the numbering is sequential and hierarchical.

Please see the attachment for further information.

Round 2

Reviewer 4 Report

Comments and Suggestions for Authors

well revised.